# Establishing an Efficient Regeneration System for Tissue Culture in *Bougainvillea buttiana* ‘Miss Manila’

**DOI:** 10.3390/plants11182372

**Published:** 2022-09-11

**Authors:** Tao Huang, Huihui Zhang, Runan Zhao, Zunling Zhu

**Affiliations:** 1Collage of Landscape Architecture, Nanjing Forestry University, Nanjing 210037, China; 2Southern Modern Forestry Collaborative Innovation Center, Nanjing Forestry University, Nanjing 210037, China; 3College of Art and Design, Nanjing Forestry University, Nanjing 210037, China

**Keywords:** *Bougainvillea*, micropropagation, plant growth regulators, rhizogenesis

## Abstract

*Bougainvillea* plants have important ornamental and cultural value, as well as applications, for example, in improving the ecological environment, sterilization and as anti-virals in medicines, etc. Among many varieties, *Bougainvillea buttiana* ‘Miss Manila’ is more popular in landscape applications in southern China because of its excellent qualities. However, because of the difficulty of sexual reproduction, low rooting rate of asexual propagation cuttings and low temperature tolerance, its large-scale development is limited. For easy, quick and mass multiplication of such cultivars, tissue culture technique can be put to use. In this study, nodal segments of ‘Miss Manila’ were used as explants, and a single-factor experiment was carried out with a variety of plant growth regulators and concentrations to establish and optimize a complete tissue culture propagation system. The results showed that the best disinfestation was 75% ethanol treatment for 0.5 min + 0.1% HgCl_2_ treatment for 9 min, and the survival rate was 86.83%. The best shoot initiation formula was MS + 6-BA (2.5 mg/L) + IBA (0.2 mg/L), and the induction rate was 95.14%. The best formula for shoot proliferation was MS + 6-BA (1.5 mg/L) + NAA (0.1 mg/L), and the proliferation coefficient was 3.13. The best rooting culture formula was 1/2 MS + IBA (2.0 mg/L), and the rooting rate was 81.97%. The survival rate for plantlet refining and transplanting was 93.33%. In this study, a relatively efficient regeneration system for the tissue culture and rapid propagation of *Bougainvillea buttiana* ‘Miss Manila’ was established to address the problems of rooting and transplanting of this variety, to benefit research on the industrialized production and regeneration systems of this variety, and to provide a reference for the tissue culture of other varieties of *Bougainvillea* or other woody plants.

## 1. Introduction

Nyctaginaceae family members are distributed in tropical and subtropical regions all over the world. At present, this family has a total of approximately 31 genera, including more than 400 kinds of herbs, subshrubs, vines, shrubs and trees [1]. *Bougainvillea*, which belongs to this family, is an important ornamental landscape plant and a flowering plant native to Brazil [2], with different bright bract colours and wide adaptability to various soil and climatic conditions, making it one of the finest and most functional ornamental plants. Later, it was introduced into Taiwan by way of the UK, and it was promoted and used in various regions, including large areas of the mainland, in the 1980s. Due to its outstanding ornamental value and environmental adaptability, there are 22 regions in Xiamen, Fujian, Yunnan and other areas that use *Bougainvillea* as a local representative flower [3], and these plants play an important role in urban landscaping in China [4].

In recent years, there have been an increasing number of studies on *Bougainvillea*, and the research has also been very extensive. In addition to using molecular biology to identify and classify the germplasm resources [5,6], many studies have focused on the gardening and production applications of *Bougainvillea*. For example, in gardening applications, *Bougainvillea* is used to regulate air [7], water [8] and soil quality [9], optimize environmental quality and reduce pollution. Moreover, it is used in pigment development and extraction [10,11], material preparation [12,13], medicinal fields [14,15], etc. Thus, its huge cultivation potential makes it desirable flowering plant in cities.

*Bougainvillea buttiana* ‘Miss Manila’ has rose-red bracts and a long flowering period. If managed properly, flowers are abundant and dense most of the year. Especially in places with sufficient light and high temperature, it is often full of flowers, resistant to pruning, and easy to sprout. The flower colour is popular among Chinese people, the flowering period is long, and the branches do not grow steeply upright, so it is easy to manage. However, due to the unique perianth tube structure, the pollen is difficult to reach the stigma, and the pollination and seed setting rate is low, resulting in low conventional sexual reproduction coefficient, long rooting cycle, and seasonal restrictions on reproduction time [16,17]. The establishment of plant tissue culture technology enables studies in broader fields of plant science, such as plant breeding, industrial production, germplasm conservation and genetic engineering. In vitro plant tissue culture can eliminate the dependence on wild plant resources and ensure that production is not affected by geographical conditions [18]. In research on the propagation of *Bougainvillea buttiana* ‘Miss Manila’, a cutting experiment showed that the rooting rate was not high [19]. Using rapid propagation via tissue culture to overcome the cross incompatibility can undoubtedly promote the production and applications of *Bougainvillea buttiana* ‘Miss Manila’.

Since the beginning of the last century, some scholars have begun to explore the establishment of a tissue culture system for *Bougainvillea* [20]. In recent years, with the wide application of *Bougainvillea*, scholars have gradually explored the aspects of explant selection, culture medium and plant growth regulator ratios [21], but there are still great differences among varieties, and there are common limitations in these studies, such as single plant growth regulator use and unreasonable experimental design. Li [22] obtained a low value-added rate and rooting rate in research on a tissue culture system of *Bougainvillea buttiana* ‘Miss Manila’ that cannot meet the demands for industrialization in China. Therefore, Thisstudy included the complete process from the establishment of an aseptic system, primary culture, subculture, rooting culture to plantlet acclimatization and transplantation. This study will provide an important technical method for the large-scale production of *Bougainvillea buttiana* ‘Miss Manila’ and promote its industrial development and lay the foundation for the establishment of an effective regeneration system.

## 2. Results

### 2.1. Disinfestation of Explants

The nodal segments were tested with different disinfestant times under the combination of 75% ethanol and 0.1% HgCl_2_. In the statistics of the nodal segment explant contamination rate, the results showed (Figure 1a) that with the increase in the treatment time of 0.1% HgCl_2_, the explant contamination rate showed a downwards trend, and the lowest contamination rate (3.49%) was reached with 75% ethanol treatment for 1 min + 0.1% HgCl_2_ treatment for 9 min. In combination with the 0.1% HgCl_2_ treatment for 6~8 min, the disinfestant effect of 75% ethanol treatment for 1 min was significantly better than that of 75% ethanol treatment for 0.5 min. In the statistics of the mortality of nodal segment explants (Figure 1b), the results showed that with the increase of the treatment time of 0.1% HgCl_2_, the mortality of explants increased and reached the highest level (10.42%) under 75% ethanol treatment for 1 min + 0.1% HgCl_2_ treatment for 9 min. In combination with the 0.1% HgCl_2_ treatment for 8~9 min, the mortality of explants treated with 75% ethanol for 1 min was significantly higher than that of those treated with 75% ethanol for 0.5 min. In the statistics for the survival rate of nodal segment explants (Figure 1c), the results showed that with the increase in the treatment time of 0.1% HgCl_2_, the survival rate of explants increased and reached the highest level (86.83%) when 75% ethanol was treated for 0.5 min + 0.1% HgCl_2_ for 9 min. With the treatment of 0.1% HgCl_2_ for 6~9 min, the survival rate of explants treated with 75% ethanol for 0.5 min changed significantly compared with that of those treated with 75% ethanol for 1 min.

### 2.2. Induction Culture

Nodal segments with axillary buds were selected as explants for the induction culture. Under culture with different plant growth regulator ratios, the axillary buds of the nodal segments began to sprout in about 7 days and then continued to elongate, showing obvious internodes. At the same time, the bottom of the nodal segment expanded, and callus formed in all parts of the nodal segment. According to the observations, when callus formed at the bottom of the nodal segment, it tended to be loose and swollen (Figure 2a). The results showed (Table 1) that under the test medium, the induction rate of the IBA treatment group with the same concentration was mostly higher than that of the NAA treatment group. Only when the concentration of 6-BA was 2.0 mg/L was the induction rate of the IBA treatment group with an auxin concentration of 0.1 mg/L lower than that of the NAA treatment group, and the highest induction rate (95.14%) was achieved under the combination of 6-BA (2.5 mg/L) and IBA (0.2 mg/L), while the lowest induction rate (32.59%) was achieved under the combination of 6-BA (1.0 mg/L) and NAA (0.1 mg/L). Under the tested medium, callus formation rate of the NAA treatment group with the same auxin concentration was higher than that of the IBA treatment group. Under the ratio of 6-BA and NAA, callus formation rates were very high, all above 97.00%, and the lowest callus formation rate (62.33%) was achieved under the combination of 6-BA (2.0 mg/L) and IBA (0.2 mg/L). Under the test medium, the maximum axillary bud length (1.57 cm) was obtained under the combination of 6-BA (2.5 mg/L) + IBA (0.2 mg/L), and the minimum axillary bud length (1.20 cm) was obtained under the combination of 6-BA (2.0 mg/L) + IBA (0.2 mg/L).

### 2.3. Proliferation Culture

The axillary buds from which the culture was initiated were selected for proliferation culture, and new buds could be extended from the axillary buds (Figure 2b). In the case of adding only cytokinin, the results showed (Figure 3) that when the concentration of the cytokinin was constant, there was a significant difference between 6-BA and KT and ZT in the proliferation of buds, while there was no significant difference between KT and ZT. Under the condition of the same cytokinin level, there was a significant difference in the proliferation of buds between different concentrations of 6-BA, but there was no significant difference in the proliferation of buds between different concentrations of KT and ZT. In the test medium, when the concentration of 6-BA was 1.5 mg/L, it reached the highest value-added rate (2.63), and when the concentration of ZT was 1.5 mg/L, it reached the lowest value-added rate (1.05).

When the concentration of 6-BA was 1.5 mg/L and auxin was added at a constant rate, there was a significant difference between NAA and IBA, while NAA and IAA as well as IBA and IAA only showed differences at specific concentrations (Figure 4). Under the same auxin conditions, the treatment concentration of NAA was significantly different at 1.0 mg/L, while the difference between IBA and IAA was not significant. In the test medium, when the concentration of 6-BA was 1.5 mg/L, the highest value-added rate (3.13) was reached when the concentration of NAA was 0.1 mg/L, and the lowest value-added rate (1.26) was reached when the concentration of IAA was 1.0 mg/L.

### 2.4. Rooting Culture

Strong single buds were transferred to the rooting medium. After 7 days, the bases began to expand slightly, and at approximately 14 days, white small roots began to protrude from the bottoms of the stems, and then the small roots gradually elongated, accompanied by new roots (Figure 2c). The results of different auxin treatments showed that (Table 2), under the same concentration of auxin, most of the different auxins had significant differences. Under the same auxin conditions, the difference in auxin concentration between 1.5~2.0 mg/L was significant. In the case of the same kind of auxin, the concentration of auxin was significantly different between 1.5~2.0 mg/L. In the test medium, when the concentration of NAA was 1.5 mg/L, the highest rooting rate (69.21%) of this group was reached (Figure 2d). When the concentration of IBA was 2.0 mg/L, the highest rooting rate (81.97%) was achieved (Figure 2e). When the concentration of IAA was 1.5 mg/L, the highest rooting rate (52.70%) was achieved (Figure 3f). In terms of root number, most of the differences between the treatments were not significant. In the test medium, when the concentration of NAA was 2.0 mg/L, the maximum root number (4.75) was reached, and when the concentration of IAA was 0.5 mg/L, the minimum root number (2.14) was reached. In terms of root length, most of the differences between the treatments were not significant. In the test medium, when the concentration of IBA was 0.5 mg/L, the maximum root length (2.76 cm) was reached, and when the concentration of IBA was 1.0 mg/L, the minimum root length (0.77 cm) was reached.

### 2.5. Plantlet Acclimatization and Transplanting

The rooted plantlets (Figure 2g) transplanted after acclimatization grew well and fast in the artificial climate chamber culture (Figure 2h). According to statistics, the average survival rate of the transplanted plantlets in the three-hole plates was 93.33%; plantlets were then transplanted into separate large pots (Figure 2i), which were transferred to the greenhouse, and grew well.

## 3. Discussion

### 3.1. Effects of Different Disinfestant Treatment on Explants

We can be sure that with the increase in 75% ethanol and 0.1% HgCl_2_ treatment time, the disinfestation of bacteria carried by explants was better achieved, resulting in the decline in the contamination rate. At the same time, the damage to the explants also increased, resulting in an increase in mortality. It is worth noting that the treatment of 75% ethanol was designed with two durations, mainly to study the influence of 75% ethanol with different treatment times on the contamination rate of explants. Previous studies have shown that appropriately increasing the duration of 75% ethanol treatment can effectively reduce the contamination rate [23]. Carbendazim and 8-HQC are also used as explant disinfestation reagents [24]. Compared with that reported with their use, this study obtained a relatively higher survival rate of explants. The main analysis shows that ethanol, as an organic solvent, has a better wetting effect on explants in addition to its bactericidal function, so it has a better bactericidal effect in combination with HgCl_2_.

### 3.2. Effects of Different Plant Growth Regulator Treatments on Induction Culture

In this experiment, the nodal segment was selected as the explant, which is representative of the type of axillary bud proliferation. It is a process from callus to organ regeneration, which reduces the probability of chromosome variation in cells and the possibility of clonal variation [25]. At the same time, axillary buds arise from axillary bud primordia, and the shoots are relatively strong, so industrial production does not require strong shoots culture [21]. Under the combination of 6-BA and NAA, the induction of axillary buds of explants was promoted with the increase in the concentration of 6-BA and NAA, indicating that explant induction may require a relatively high concentration of 6-BA and NAA, which is consistent with the research results of Zhang [26]. Under the combination of 6-BA and IBA, when the concentration of 6-BA was 2.0 mg/L, the increase in IBA concentration promoted the induction of explants, while when the concentration of IBA was 0.1 mg/L, the increase in 6-BA concentration inhibited the induction of explants, indicating that the induction of explants may require a relatively low combination of 6-BA and a relatively high concentration of IBA. Contrary to the test results of Du [27], there were differences in the response to plant growth regulator induction between different species. In addition, Datta [28] obtained the best induction response on culture medium supplemented with 2.0 mg/L BA + 0.5 mg/L NAA + 0.5 mg/L GA_3_ in the experiment of two cultivated varieties, ‘Los Banos variegata’ and ‘Mary Palmer special’, and the induction rate was as high as 1, considering the differences in the ability of bud induction ability among species of *Bougainvillea*.

It is worth noting that in the study of callus induction in *Bougainvillea glabra*, Sun [29] found that the nodal segment, as an explant, was significantly better than the leaves and petioles in terms of callus induction. This experiment also showed the high callus formation rate of the nodal segment, indicating that the nodal segment is a good material for studying the healing response of *Bougainvillea*. However, in this experiment, callus obtained did not differentiate on the test medium, and gradually browning and even death occurred (Figure 5), so no follow-up discussion occurred. It may be that callus may be composed of different cell types, and only some cells may participate in organ regeneration [30], resulting in the difficulty of differentiation. Gong [31] induced and differentiated callus of *Bougainvillea glabra* ‘Sanderiana’ with MS + 3.0 mg/LBA + 0.2 mg/L 2,4-d + 0.1 mg/LNAA. It was considered that the failure to differentiate was caused by a variety of differences and inappropriate media and plant growth regulators. The research group also selected leaves as explants, which had accumulated a certain amount of callus induction, but further experiments focusing on callus differentiation are still needed. If a regeneration system can be successfully established, it will greatly improve the proliferation coefficient and lay the foundation for a genetic transformation system for *Bougainvillea*.

### 3.3. Effects of Different Plant Growth Regulator Combinations on Proliferation Culture

The use of cytokinins can eliminate apical dominance, thereby promoting the formation of additional buds, and the type and concentration of cytokinins will affect the proliferation factor [32]. Among the three cytokinins tested, 6-BA was significantly better than ZT and KT in terms of proliferation, indicating that *Bougainvillea* had a better response to 6-BA in terms of proliferation. Research shows that the appropriate combination of cytokinin and auxin is conducive to providing a higher proliferation coefficient [33]. When auxin was added to 1.5 mg/L 6-BA, the proliferation coefficient first increased and then decreased with increasing concentration. The addition of NAA at a concentration of 0.1 mg/L had a significant promoting effect on the proliferation coefficient. In the proliferation of *Bougainvillea spectabilis* by Shah [34], the highest proliferation coefficient reached 5.41, while the highest proliferation coefficient in this test was 3.13, which may have been due to the differences between varieties or the influence of subculture times on the proliferation coefficient [35]. The test materials were subcultured more than 8 times. Compared with that of the initial subculture, the proliferation coefficient was significantly reduced. In addition, this test takes a robust single bud of more than 1.0 cm as the statistical sample, which may be different from other sample selection standards for the statistical analysis of multiplication coefficients.

### 3.4. Effects of Different Plant Growth Regulator Combination Combinations on Rooting Culture

The root formation process of tissue-cultured seedlings is a series of physiological and molecular processes stimulated by auxin, but different auxins have different root induction abilities, which are related to the different affinities of auxin receptors involved in the process of rhizome formation [36] and to the different concentrations of free auxin reaching target cells [37]. Therefore, suitable auxin species and concentrations were selected according to different plants to regulate the formation of roots of tissue-cultured seedlings. In this experiment, 2.0 mg/L IBA resulted in the highest rooting rate (81.97%), which was much higher than that of Du (39%) [27]. Pratiksha [38] found that the highest rooting rate obtained by IBA was higher than that of NAA when rooting and culturing two difficult rooting varieties, which was consistent with the results of this study. Compared with Shah’s 2.5 mg/L NAA + 2.5 mg/L IBA, a 100% rooting rate of *Bougainvillea spectabilis* tissue-cultured seedlings was obtained, considering the difference in rooting ability among varieties. At the same time, some experiments showed that the rooting effect of mixed use of multiple auxins was better than that of single auxins [39]. This experiment was mainly aimed at screening different concentrations of the same auxins. In follow-up studies, we can continue to explore the influence of the mixed use of different auxins on the rooting culture of *Bougainvillea*.

## 4. Conclusions

To establish and optimize an efficient regeneration system for tissue culture, a single-factor experiment was carried out with *Bougainvillea buttiana* ‘Miss Manila’as the material under a variety of plant growth regulators types and concentrations. Firstly, pretreat the explants (nodal segments) with 75% ethanol treatment for 0.5 min + 0.1% HgCl_2_ treatment for 9 min, and culture them on MS media supplemented with 6-BA (2.5 mg/L) + IBA (0.2 mg/L); followed by shoot proliferation on MS + 6-BA (1.5 mg/L) + NAA (0.1 mg/L). For rooting culture the shoot cultured on 1/2 MS + IBA (2.0 mg/L) and then acclimatized in two steps. In general, a relatively efficient regeneration system for tissue culture was herein established. Compared with previous studies, great progress has been made in the development of sterile systems, proliferation culture, refining and transplanting. However, in terms of further breakthroughs in the proliferation coefficient, the system can be further explored by adjusting the components of the basic medium, using new plant growth regulators, or exploring factors such as light quality.

## 5. Materials and Methods

### 5.1. Experimental Materials

The experiment was carried out in the Jiangsu Provincial Key Laboratory of Landscape Architecture, Nanjing Forestry University, from 2021 to 2022 (E: 118°49′12″, N: 32°4′12″). The seedlings were purchased from Zhangzhou, Fujian Province, China, in March 2021 with cuttings with a height of 20~30 cm and then placed in the greenhouse of the National Experimental Teaching Demonstration Center for Landscape Architecture, Nanjing Forestry University, for slow seedling cultivation (Figure 6). After the growth state became consistent and good, the experiment began in mid-April. All plant plant growth regulators including 6-BA (6-benzyladenine, available ingredient 99%), KT (kinetin, 99%), ZT (zeatin, 99%), NAA (α-naphthaleneacetic acid, 98%), IBA (indole-3-butyric acid, 99%) and IAA (indole-3-acetic acid, 99%) were provided by the Shanghai Yuanye Biotechnology Co., Ltd. (Shanghai, China), as are sucrose (98%) and agar (strength 1200 g/cm^2^).

### 5.2. Experimental Design

#### 5.2.1. Explant Disinfestation

On a sunny morning, the sprouting branches of the current year were collected from plants with vigorous growth that were free of diseases and pests, and the collected branches were immediately brought back to the laboratory for cutting. Small nodal segments with a length of approximately 3.0 cm were cut, ensuring that there were 1~2 buds on each segment; the cut explants were put into pure water with laundry detergent and stirred constantly to produce rich foam. After soaking for 30 min, the samples were sealed with gauze and rinsed under running water for 2 h.

The washed explants were transferred to laminar air flow bench, first treated with 75% ethanol, then transferred to 0.1% HgCl_2_, and then rinsed with sterile distilled water more than 5 times. The treated explants were placed on sterilized filter paper to remove excess water for later treatments. Eight different treatment time combinations of 75% ethanol and 0.1% HgCl_2_ were designed, including 75% ethanol (0.5 min) + 0.1% HgCl_2_ (6, 7, 8, 9 min) and 75% ethanol (1 min) + 0.1% HgCl_2_ (6, 7, 8, 9 min), to select the best disinfestant treatment time for explants. It is worth noting that HgCl_2_ is toxic as a heavy metal salt bactericide. After use, it can be filtered by qualitative filter paper and used next time. If you want to discard it, please contact the official laboratory waste liquid treatment company for treatment.

#### 5.2.2. Culture Initiation

On the laminar air flow bench, both ends of the explant were cut off with a scalpel, and the small segment with axillary buds was transferred to the test initiation medium. As a result of the preliminary experiment, it is found that the combination of cytokinin and auxin is more likely to induce buds, so MS medium was selected as the basic medium for primary culture, and the plant growth regulators used included 8 combinations of 6-BA, NAA and IBA: (1) 1.0 mg/L 6-BA + 0.1 mg/L NAA, (2) 2.0 mg/L 6-BA + 0.1 mg/L NAA, (3) 2.0 mg/L 6-BA + 0.2 mg/L NAA, (4) 2.5 mg/L 6-BA + 0.2 mg/L NAA, (5) 1.0 mg/L 6-BA + 0.1 mg/L IBA, (6) 2.0 mg/L 6-BA + 0.1 mg/L IBA, (7) 2.0 mg/L 6-BA + 0.1 mg/L IBA, and (8) 2.5 mg/L 6-BA + 0.1 mg/L IBA to determine the optimal plant growth regulator type and concentration for culture initiation.

#### 5.2.3. Proliferation Culture

Robust single axillary buds with a length of more than 1.0 cm obtained from the culture initiation were cut as experimental materials and transferred to proliferation medium for proliferation culture. In this stage, cytokinin mainly plays a role, and the purpose of adding auxin is to seek a breakthrough in the proliferation coefficient, and the shoots with 4–5 times of subculture were selected for the experiment. The test was divided into two steps:
(I)Different concentrations of cytokinins were added to MS medium, and the effects of different cytokinins, including different concentrations of 6-BA, KT and ZT, on the proliferation and growth of shoots in test tubes were analysed and compared. A total of 12 combinations, four concentrations of each of the three cytokinins were treated separately, including6-BA (0.5, 1.0, 1.5, 2.0 mg/L), KT (0.5, 1.0, 1.5, 2.0 mg/L), and ZT (0.5, 1.0, 1.5, 2.0 mg/L), were selected. Thus, the optimal type and concentration of cytokinin in proliferation culture were selected.(II)Based on the optimal treatment of the cytokinin in (I), MS was used as the basic medium to analyse and compare the effects of auxin combinations on the proliferation and growth of test tube shoots. NAA, IBA and IAA were added at different concentrations, four concentrations of each of the three auxins were treated separately, including NAA (0.05, 0.1, 0.5, 1.0 mg/L), IBA (0.05, 0.1, 0.5, 1.0 mg/L), and IAA (0.05, 0.1, 0.5, 1.0 mg/L), to select the most suitable plant growth regulator type and concentration in proliferation culture.

#### 5.2.4. Rooting Culture

The single buds obtained at the end of subculture were connected to the rooting medium with 1/2 MS as the basic medium, and the effects of auxin combinations on the rooting and growth of shoots were analysed and compared. NAA, IBA and IAA were added at different concentrations, for a total of 12 combinations, including NAA (0.5, 1.0, 1.5, 2.0 mg/L), IBA (0.5, 1.0, 1.5, 2.0 mg/L), and IAA (0.5, 1.0, 1.5, 2.0 mg/L). Thus, the most suitable rooting medium was selected.

#### 5.2.5. Plantlet Acclimatization and Transplanting

Single shoots that were successfully rooted with good root growth in tissue culture were cultivated and transplanted. First, the sealing film was opened and plantlets together with tissue culture vessel were placed in the artificial climate chamber for 3 days to gradually reduce the air humidity and enhance the light. Then, the residual root medium was removed and washed with water, and plants were transplanted to a 50-hole cell tray (each single hole was 5 × 5 × 10 cm) with mixed organic cultivation substrate of vermiculite, perlite and peat after high-pressure sterilization (patent number: zl201310709304.0), and the cell trays were placed in the artificial incubator after transplantation. The cultivation process in the artificial climate incubator included a two-step method. First, the rooted plantlets planted in the substrate were covered with plastic fresh-keeping film, and the light was controlled within 2000 Lux. After one week, the light intensity was gradually increased to 6000 Lux, and the fresh-keeping film was opened. The photoperiod was 12 h/d, the temperature was 25 ± 2 °C, and the humidity was more than 85%.

### 5.3. Determination of Morphological Indices

For explant disinfestation, culture initiation, proliferation culture and rooting culture, 30 explants were inserted per treatment. In plantlet acclimatization and transplanting stage, 50 plants were transplanted. The culture medium inserted with explants, with a sucrose concentration of 3% and agar concentration of 0.7%, pH 5.8~6.0, was maintained in the culture room (with controlled temperature and illumination time). The photoperiod was 12 h/d, the temperature was 25 ± 2 °C, and the light intensity was set to 2000 Lux. The differentiation, proliferation and rooting of plants were observed every day, and the statistical cycle was set as 30 days. During the establishment stage of explant disinfestation, the contamination rate and mortality rate were counted. In the culture initiation stage, the initiation rate, callus formation rate and axillary bud length were determined. In the proliferation culture stage, the proliferation coefficient was determined. At the rooting culture stage, the rooting rate, root length and root number were counted.

### 5.4. Statistical Analysis

The experiments conducted according to completely randomized block design (RBD) for single factor experiments with triplicates for each treatment. The data were subjected to analysis of variance and the significance of differences among mean values was carried out using Duncan’s Multiple Range Test (DMRT) at *p* < 0.05 using SPSS software, version 24.0. The results were expressed as mean ± standard deviation (SD) of triplicates.

## Figures and Tables

**Figure 1 plants-11-02372-f001:**
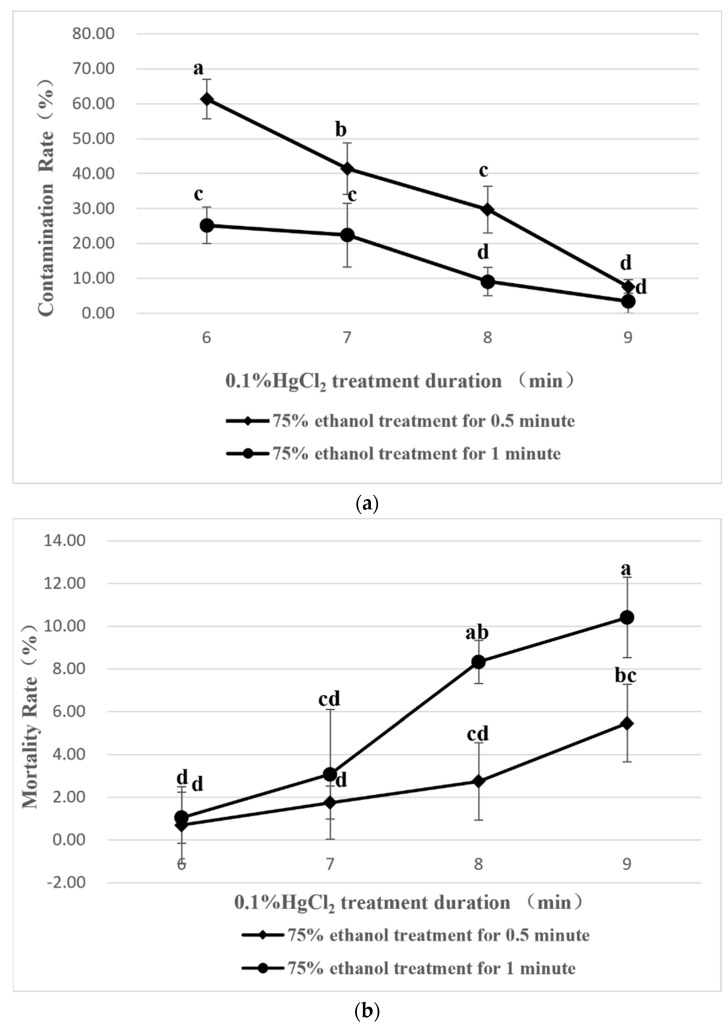
Effects of ethanol and HgCl_2_ on explants disinfestation: (**a**) Effects of different disinfestant treatments on the contamination rate; (**b**) Effects of different disinfestant treatments on the mortality rate; (**c**) Effects of different disinfestant treatments on the survival rate. The different letters indicate significant differences among eight ethanol and HgCl_2_ treatment combinations (*p* < 0.05).

**Figure 2 plants-11-02372-f002:**
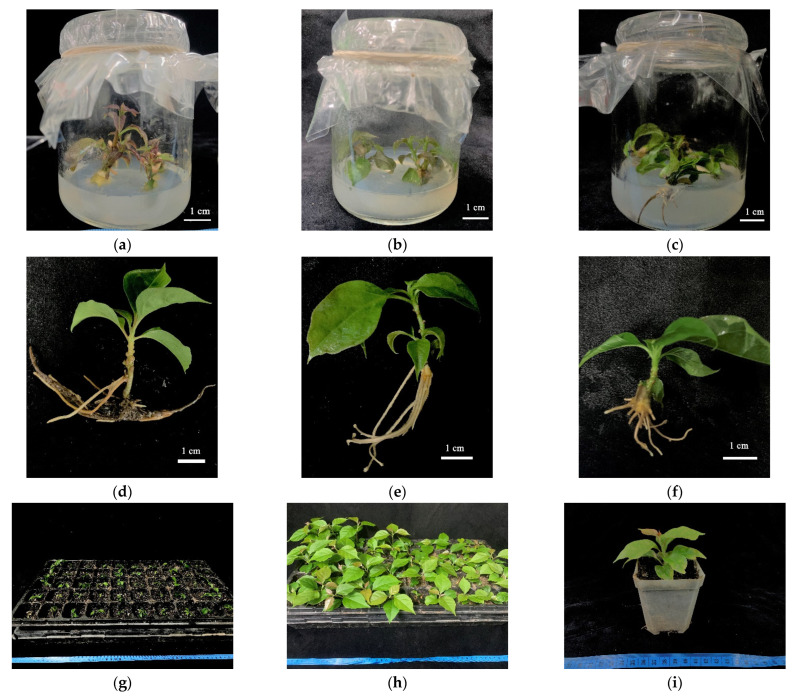
Diagram of each stage of tissue culture and propagation of *Bougainvillea buttiana* ‘Miss Manila’: (**a**) Culture initiation; (**b**) Proliferation culture; (**c**) Rooting culture; (**d**) 1.5 mg/L NAA treatment in rooting culture; (**e**) 2.0 mg/L IBA treatment in rooting culture; (**f**) 1.5 mg/L IAA treatment in rooting culture; (**g**) Plantlet acclimatization and transplanting; (**h**) 3-week incubation in an artificial climate chamber; (**i**) Pot changing and transplanting diagram.

**Figure 3 plants-11-02372-f003:**
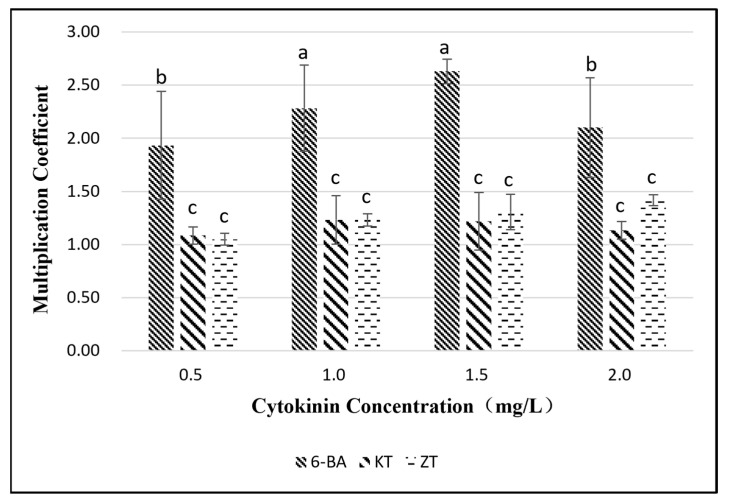
Effect of 6-BA, KT and ZT on shoot proliferation of *Bougainvillea buttiana* ‘Miss Manila’. The effect of 6-BA treatment group was the best, and when its concentration was 1.5 mg/L, the maximum proliferation coefficient is reached. The different letters indicate significant differences among twelve cytokinin species and concentration treatments in proliferation culture (*p* < 0.05).

**Figure 4 plants-11-02372-f004:**
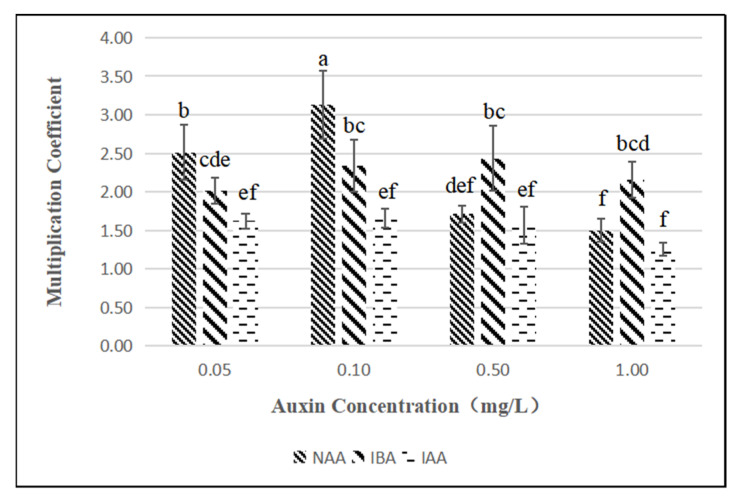
Effect of NAA, IBA and IAA on shoot proliferation of *Bougainvillea buttiana* ‘Miss Manila’. The effect of NAA treatment group was better, and when its concentration was 0.1 mg/l, the maximum proliferation coefficient is reached. The different letters indicate significant differences among twelve auxin species and concentration treatments in proliferation culture (*p* < 0.05).

**Figure 5 plants-11-02372-f005:**
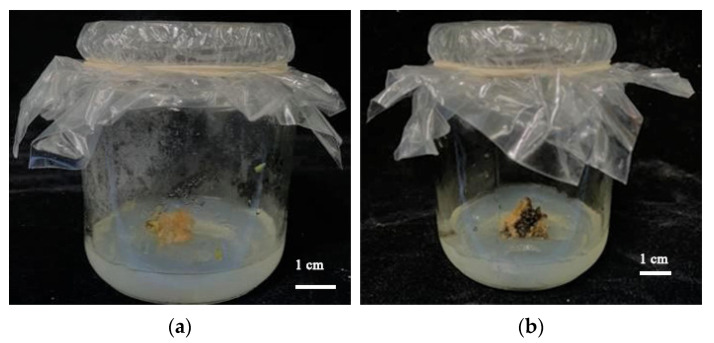
Changes of callus during callus culture: (**a**) culture for one week; (**b**) culture for one month.

**Figure 6 plants-11-02372-f006:**
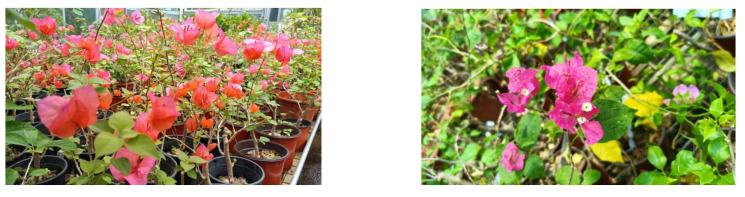
*Bougainvillea buttiana* ‘Miss Manila’.

**Table 1 plants-11-02372-t001:** Effect of 6-BA and NAA, IBA on axillary bud induction of *Bougainvillea buttiana* ‘Miss Manila’.

Treatment Group	Initiation Rate (%)	Callus Induction Rate (%)	Shoot Length (cm)
6-BA (1.0 mg/L) + NAA (0.1 mg/L)	32.59 ± 4.39 d	98.00 ± 3.46 ab	1.29 ± 0.19 ab
6-BA (2.0 mg/L) + NAA (0.1 mg/L)	58.19 ± 14.94 b	1.00 ± 0.00 a	1.33 ± 0.30 ab
6-BA (2.0 mg/L) + NAA (0.2 mg/L)	62.22 ± 22.34 bc	97.67 ± 4.04 ab	1.26 ± 0.18 a
6-BA (2.5 mg/L) + NAA (0.2 mg/L)	78.16 ± 8.25 ab	1.00 ± 0.00 a	1.52 ± 0.14 ab
6-BA (1.0 mg/L) + IBA (0.1 mg/L)	90.28 ± 6.78 a	90.00 ± 5.29 bc	1.27 ± 0.18 ab
6-BA (2.0 mg/L) + IBA (0.1 mg/L)	52.78 ± 2.41 c	86.33 ± 2.89 c	1.32 ± 0.14 ab
6-BA (2.0 mg/L) + IBA (0.2 mg/L)	90.14 ± 0.58 a	62.33 ± 5.51 d	1.20 ± 0.19 b
6-BA (2.5 mg/L) + IBA (0.2 mg/L)	95.14 ± 4.34 a	92.00 ± 7.81 abc	1.57 ± 0.03 ab

The different letters indicate significant differences among eight 6-BA and NAA, IBA treatment combinations (*p* < 0.05).

**Table 2 plants-11-02372-t002:** Effect of NAA, IBA and IAA on rooting responses of *Bougainvillea buttiana* ‘Miss Manila’.

Treatment Group	Rooting Rate (%)	Number of Roots	Root Length (cm)
1/2MS + NAA (0.5 mg/L)	56.01 ± 3.48 cd	2.68 ± 0.25 bcd	0.91 ± 0.13 b
1/2MS + NAA (1.0 mg/L)	63.51 ± 9.28 bc	2.93 ± 0.31 bcd	0.95 ± 0.12 b
1/2MS + NAA (1.5 mg/L)	69.21 ± 7.86 b	3.73 ± 0.42 b	1.02 ± 0.14 b
1/2MS + NAA (2.0 mg/L)	41.03 ± 8.00 ef	4.75 ± 0.94 a	0.96 ± 0.10 b
1/2MS + IBA (0.5 mg/L)	41.54 ± 7.80 ef	2.30 ± 0.26 cd	2.76 ± 0.51 a
1/2MS + IBA (1.0 mg/L)	45.59 ± 6.87 def	2.76 ± 0.57 bcd	0.77 ± 0.16 b
1/2MS + IBA (1.5 mg/L)	54.00 ± 5.29 cd	3.17 ± 0.16 bcd	1.06 ± 0.54 b
1/2MS + IBA (2.0 mg/L)	81.97 ± 7.26 a	3.32 ± 0.87 bc	1.14 ± 0.35 b
1/2MS + IAA (0.5 mg/L)	36.59 ± 1.57 fg	2.14 ± 0.23 d	1.04 ± 0.37 b
1/2MS + IAA (1.0 mg/L)	25.79 ± 5.32 gh	2.70 ± 0.50 bcd	0.94 ± 0.17 b
1/2MS + IAA (1.5 mg/L)	52.70 ± 8.61 cde	2.50 ± 0.17 cd	1.12 ± 0.47 b
1/2MS + IAA (2.0 mg/L)	20.28 ± 5.21 h	3.07 ± 1.10 bcd	0.82 ± 0.41 b

The different letters indicate significant differences among twelve auxin species and concentration treatments in rooting culture (*p* < 0.05).

## Data Availability

The data presented in this study are available on request from the corresponding author. The data are not publicly available due to the individual invention patent related to this result is being applied for.

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
