# Peer review of "Establishing an Efficient Regeneration System for Tissue Culture in Bougainvillea buttiana ‘Miss Manila’"

_plants, 2022, doi:10.3390/plants11182372_

Round 1
Reviewer 1 Report
The ornamental values of bougaivillea in urban and garden landscapes are poorly documented. Other plant characteristics are enlisted but a review on phytochemical properties of the genus is missing in this context (for example, Hindawi Evidence-Based Complementary and Alternative Medicine Volume 2018, Article ID 9070927, https://doi.org/10.1155/2018/9070927: Review Article Bougainvillea Genus: A Review on Phytochemistry, Pharmacology, and Toxicology. Rodolfo Abarca-Vargas and Vera L. Petricevich.
In discussion the following paper of Pratiksha et al. 2016 In vitro Sterilization, Rooting and Acclimatization of Difficult-to-root Bougainvillea Cultivars International Journal of Bioresource and Stress Management, 7(3):412-4190) is missing.
The paper is well written, however, the terms such as domestication and insemination apply rather to animal science and should be replaced by acclimatization and pollination/fertilization. Also „inoculation” applies to microbiology so „Strong single buds were inoculated into the rooting medium” (verse 282) is not correct. They were „placed, transferred to, inserted.
Minor technical errors:
verse 65, 69 – in vitro and via – italics;
verse 80 – the abbreviations (like KT, ZT) should be explained or a separate list of all the abbreviations attached;
verse 95-99 – spaces between the words;
verse 150,242,307 – standardize the font;
The description of statistics under table 2 should be replaced by: As in Table 1;
The same applies to the descriptions under figures 3-5
A mistake in the letter description of the photos in Fig. 3 should be corrected: (i) is under the last photo while (g) is in the text.
Cultivar names are written with capital letters – correct the verse 357 and 393.
Photos in Fig. 6 are too dark.
Reviewer 2 Report
In this manuscript written by Huang and coauthors, a relatively efficient regeneration system for the tissue culture and rapid propagation of B. buttiana 'Miss Manila' was establishe. The work was carried out by adequate methods and the results obtained are valuable. However, the work can be published after major revisions only.
The main points:
The “Conclusions” section is too general and does not contain concrete achievements - recommendations on nutrient composition/growth conditions. Reading the entire manuscript it is difficult to find out which medium/method was the best.
The Conclusions should be revised.
The words in the Keywords section should be different from those in the title. This article is repetition and should be changed here.
The authors used abbreviations of plant growth regulators that differed from commonly used (e.g. 6-BA). All shortcuts must be collected in one section or explained after first use.
The work requires linguistic correction carried out by a specialist familiar with the terminology of plant in vitro cultures. The work uses numerous terms not usually used in this type of work, for example:
· mitogen concentration (????) instead of cytokinin content or concentration
· proliferation, rooting … cultures instead proliferation, rooting stage
· domestication and transplanting instead of acclimatization
· insemination (usually used for animals) instead of fertilization
· induction instead of initiation.
These are just a few examples, there are more at work.
All tables and figures (descriptions) should be redrafted to make them more readable. They currently contain too many repetitions.
In the caption of the graphs and tables, remove the information "MS medium supplemented with 3% sucrose" - the sucrose content is the same for all media, so it makes no sense to provide it.
Change caption Fig 1 - delete Panoramas and Close up information
Improve the readability of tables: 1 and 2. Delete the repeating words: mg / l (tab. 1) and 1/2 MS (tab. 2).
After a major revision, I believe the paper deserves a Plants publication.
Reviewer 3 Report
The manuscript describes experiments to define a protocol for micropropagation of one cultivar of Bougainvillea. Introduction is very limited regarding the tissue culture literature on Bougainvillea, mentioning two articles and very little about their results. While it is an important ornamental plant, the experimentation is simple, testing variations in concentration of cytokinins, followed by auxins (apparently no combination of them were used (according to M&M), which might favor morphogenesis (balance between citokinin:auxin), however Table 1 shows combination of BAP and auxins. This needs to be better explained in M&M). Introduction did not state the current status of Bougainville tissue culture, or the difficulties found that led to this work. Improvements are needed in the Introduction, Justification, M&M and Results, as well as in figure presentations, as mentioned below. The indication to use HgCl2 for explant disinfestation, if really needed, should come with an alert of how toxic this is and how it should be handled and discarded. Other agents should be tested instead of HgCl2, this chemical should not be used in a Plant Tissue Culture Lab.
Check the term hormones throughout the text. BAP, for instance, is not an hormone (produced by plants), it is a synthetic plant growth regulator with cytokinin effect. You should use plant grow regulators throughout the text. Other terms are mentioned below, which I suggest revision throughout the text. I suggest a thorough revision of the text by a native speaker who is familiar with plant tissue culture terms in English.
Lines 15-17: “..... due to the pollen tube structure of Bougainvillea, the sexual reproduction rate is low, while the low rooting rate of cuttings in asexual reproduction and the breeding environment restrict its large-scale development.”
Comments- It seems the authors want to emphasize difficulties with reproduction and rooting in Bougainvillea, affecting plant propagation? This is just an abstract, so I believe there is no need to be that specific, mentioning that the pollen tube structure causes a low reproduction rate. If so, I would suggest being even more specific and saying what in its structure is the cause for this. But I would do so in the introduction and not in the abstract. On the other hand, I don’t think this manuscript addresses sexual reproduction, but only micropropagation. Introduction/justification in the abstract needs improvement.
What do the authors mean by “...breeding environment restrict its large-scale development”?
Lines 16-17 - The objectives of the work are not mentioned in the abstract. From the introduction the authors already describe materials and methods.
Lines 13 and 48-53: are all these uses mentioned, besides the ornamental uses, economically significant to Bougainvillea as they sound, or these are a potential of the culture?
Lines 20-21 and 112/114/115: was HgCl2 the only option for disinfestation? It is a rather toxic agent and it should not be stimulated, so I would suggest that this is mentioned in the manuscript together with the precautionary measures that should be taken in its handling, and recommendation of use only in cases in which all other options were unsuccessful. If preliminary tests were done with other products to define that HgCl2 is needed, please mention them, to justify its use in this case.
Line 31 - Keywords should be more specific to what this manuscript itself presents and not general terms. Words that are present in the manuscript title should not be repeated as a keyword.
Line 54 - I would suggest writing the genus in full when starting a new sentence and not abbreviated.
Line 59 - What is a “flower tube”?
Line 60 - “insemination” is a term more appropriate for humans, or animals. Do you mean “pollination”? Or “fertilization”? What do the authors mean by “other reasons”?
Lines 64/65 - “Germplasm protection” could be better expressed as “germplasm conservation”. By “plant protection” are you referring to “plant germplasm conservation” also?
Line 73: What do the authors mean by “domestic scholars”?
Lines 80-81: write the plant growth regulators in full in the first time you mention them.
Line 98: I suggest to use the past tense, to make it uniform: “... were provided...”
Line 100: Figure 1 should have a better caption. Please explain what the authors mean to show with these figures.
Item 222. I believe you mean “Culture initiation”, instead of “Initiation Culture”
Items 2.2.2 and 2.2.3 do not mention for how long cultures were kept, nor if subcultures were done. Subculture is mentioned in line 144, but not explaining how many subcultures were done in the previous tests.
Item 2.2.3: From which earlier treatments were these axillary buds obtained? Or these were obtained from the greenhouse? Please, clarify.
Line 108: what is a washing powder? Is this a laundry detergent?
Line 113: where were the treated explants placed? Under the hood?
Line 116: it should be disinfestant treatment, or disintestation, instead of “disinfectant”. Line 185: disinfestation, instead of disinfection. Please check throughout the text.
Line 118: Was this performed under the hood, or on a clean bench in the lab? In tissue culture the explants are introduced in vitro under the aseptic hood, so we assume this, however, here the clean bench, not hood, is specified, so it is not clear.
Line 133: “seedlings” would be the plants obtained from seed germination, which is not the case here. Revise “seedling” throughout the manuscript.
Lines 134-135: Combinations? There are 4 concentrations of each of 3 citokinins. Were these citokinins combined? How?
Line 147: combinations are mentioned, but with 3 auxins x 4 concentrations. Were these 3 auxins combined? How?
Line 150: again “seedlings”, term used for plants derived from seeds, not the case here. Plantlets can be used. What do you mean by “domestication”? Shouldn’t it be acclimatization?
Line 167: inoculation shall be used for introducing bacteria, fungi or virus in an experiment. For plant tissue culture the term “introduction of explants” is preferred.
Line 168: I suggest ....”was maintained in the culture room...”, instead of “stored”
Line 169: instead of “light time”, I suggest “photoperiod”
Line 205: Figure 2a: Y axis should be “contamination rate”. Explants should be “disinfested”, not “sterilized”. Check terms throughout the text.
Line 209: should be “ethanol”, not “alcohol”
Line 219: ....”stem segment expanded, and callus formed...” (remove “the” before “callus”. Check throughout the text.
Line 237: Figure 3 should be organized as a plate, with letters in the individual figures and bars within the figure should indicate the size and not the blue measuring tape for this purpose. The legend should be self explanatory, which means not just writing what each individual figure shows, but why it is shown, what do the authors would like to point out on each individual figure? In the footnote, write in full all the abbreviations. If needed size bars should be drawn in each individual Figure, instead of using a measuring tape.
Line 242: Table 1 - Use Bougainvillea in full, not abbreviated
Line 249 - what is “the bottom of the axillary buds”? Figures need to have better legends.
Line 260: Figure legend must be self-explanatory, so the reader can read a Figure without having to read the text.
Line 308: what do you mean by “refining seedlings”? Please, review the term seedling, as mentioned earlier.
Lines 451 and 457: references 12 and 14 are the same. Delete one of them in te references and correct the numbering throughout the text.
Line 468 - Check the last names of the authors of this reference, which are: Espinosa‑Leal, C.A.; Puente‑Garza, C.A.; García‑Lara, S.
Line 507 - Missing letters in the title of this reference “.... from ex vitro explants.” Access to the full article in: http://scielo.sld.cu/scielo.php?script=sci_abstract&pid=S2074-86472019000100053&lng=en&nrm=iso&tlng=en
Round 2
Reviewer 2 Report
I have read the revised manuscript and believe that the authors have made all the changes I have suggested. At present, the work is written correctly and fully deserves publication in the Plants journal.
Author Response
Thank you very much for your valuable comments on my manuscript, which are very helpful to me. Thank you sincerely again!
Reviewer 3 Report
The autors responded to all the comments and made changes to the manuscript accordingly.
Author Response

(The authors gave the same response as above.)
